# An IoT-Based Solution for Intelligent Farming [note 1]

**DOI:** 10.3390/s19030603

**Published:** 2019-01-31

**Authors:** Luís Nóbrega, Pedro Gonçalves, Paulo Pedreiras, José Pereira

**Affiliations:** 1DETI/IT—Universidade de Aveiro, 3810-193 Aveiro, Portugal; pbrp@ua.pt; 2ESTGA/IT—Universidade de Aveiro, 3810-193 Aveiro, Portugal; pasg@ua.pt; 3iFarmTec, 3680-264 São João da Serra, Portugal; jose@ifarmtec.pt

**Keywords:** IoT, M2M stack, constrained networks, intelligent farming, IoT gateway

## Abstract

Intelligent farming is one of the vast range of applications covered by the Internet of Things concept. Notwithstanding, such applications present specific requirements and constraints that are dependent on their purpose. A practical case on which that is particularly relevant is the SheepIT project, where an automated IoT-based system controls grazing sheep within vineyards, guaranteeing that they do not threaten cultures. Due to its rigid requirements, particularly regarding the deployment of the Wireless Sensor Network, Machine-2-Machine communications and necessary interactions with a computational platform available through the Internet, Internet Protocol-based solutions are not suitable. Consequently, a customized communication stack has been developed, that intends to meet the project requirements, from the physical to the Application Layers. Although it has been developed considering the SheepIT requirements, its use may be extended to more generic intelligent farming applications, since most of the requirements are directly related with the farming environment. This paper reviews the proposed stack and details the recent developments. Particularly, we focused on Internet of Things/Machine-2-Machine interaction, describing the design and deployment of a gateway that addresses the SheepIT service requirements. Additionally, and complementary to previously published results, we evaluate the gateway performance and show its feasibility and scalability in a real scenario.

## 1. Introduction and Motivation

The massification and exponential proliferation of the number of electronic devices connected to the Internet triggered the emergence of the Internet of Things (IoT) concept. With it, we are moving towards an *anytime, anyone, anything* connectivity paradigm [1], with applications for different activity sectors, such as transportation, smart things (e.g., smart cities, smart domotics, smart health), e-governance, assisted living, e-education, retail, logistics, automation, industrial manufacturing and naturally, intelligent farming [2].

Focusing on the intelligent farming domain, it is consensual that the employment of IoT-based applications can play an important role in the sustainability of the sector [3]. On the one hand, it helps farmers on their daily tasks. On the other hand, it may contribute to increasing the productivity and sustainability of the sector, through the optimization of farming processes. Notwithstanding, such application domain is surrounded by constraints, either associated with technologies or to the specificities of farmings’ users. The former include limitations and trade-offs between energy consumption, device’s size/weight, autonomy and wireless communication coverage [4]. The latter is related to the intrinsic mistrust of farmers when adopting new technologies [5].

When applying IoT to the agriculture domain, Machine-to-Machine (M2M) communications are a central key, enabling ubiquitous and autonomous communications among multiple devices without human mediation [6]. Thus, requirements such as efficiency, scalability, low-cost hardware and low-power consumption are transposed to this kind of communications. However, these requirements are dramatically dependent on the application scenarios, which has led to the emergence of a vast number of protocols, standards, communication stacks and architectures, each one addressing specific subsets of issues and/or applications.

Commonly, these architectures are structured as protocol stacks with different numbers of layers [7], enabling levels of abstraction between them and the use of different protocols. The simplest one is based on three layers: the Perception Layer, the Network Layer and the Application Layer. However, as this architecture is considered too simplistic, other models with additional layers were proposed [8]. Albeit the most trendy one is based on five layers, practical IoT systems implementations commonly use four layers (Physical and MAC Layer, Network Layer, Transport Layer and Application Layer) [9], following an IP-based network architecture. Our proposal is based on a four layer communication stack, and thus, in this paper the most relevant protocols on each one of these layers are presented.

Succinctly, IEEE.802.15.4 [10] is one of the most popular technologies used on the Physical and MAC layers. At the Network layer, the Internet Protocol (IP), either IPv4 and IPv6, is generally used, eventually with adaptation layers (e.g., 6LoWPAN [11]) to deal with resource-constrained devices. Finally, at the Application Layer, solutions such as the Constrained Application Protocol (CoAP) [12], the Advanced Message Queuing Protocol (AMQP) [13], the Message Queuing Telemetry Transport (MQTT) [14] and the Hypertext Transfer Protocol (HTTP) [15] are commonly found.

However, several other approaches coexist, due to the high heterogeneity of this kind of application and increasing need for low-cost and low-power solutions. SIGFOX [16], NB-IOT [17], LTE-M [18] and LoRa [19] are some popular examples that are part of the Low-Power Wide-Area Network (LPWAN) technologies. These protocols share the low energy consumption at expenses of limited functionality and bandwidth.

This heterogeneity of solutions also led to the development of several IoT gateways (hardware and standards) [20] to allow the interaction between different networks, protocols and communication technologies. Here, most research efforts are focused on the horizontal integration of different protocols and communication technologies, leaving aside additional services that may be relevant for some IoT applications.

Despite the significant research effort devoted to M2M/IoT applications in recent years, there are still practical cases that exhibit requirements and constraints that are not addressed nor satisfied by existing solutions. One of these applications is addressed by the SheepIT project [21], which aims at deploying an IoT-based solution for monitoring, condition and managing flocks of sheep, such that animals could weed vineyard cultures (or similar ones) in an environmentally-friendly way.

This application shares many of the common IoT features and requirements. For instance:The amount of data exchanged is relatively low and size limited;Devices shall have low size and weight;To avoid constant battery replacements and/or recharges, a high autonomy is required;The geographical area to be covered can be relatively vast;The system shall be robust and reliable.

Additionally, SheepIT exhibits a few distinctive requirements that, when combined with the aforementioned ones, preclude the use of existing IoT solutions. The complete and detailed set of requirements can be found in [22], but they can be briefly summarized as follows:The presence of irregular slopes with several obstacles;Unforeseeable node mobility (sheep);Variable node densities due to the flock movement;Need for real-time localization, preferably resorting to Received Signal Strength Indicator (RSSI)-based localization;Need to ensure the coexistence of an effective control posture mechanism;Frequent system reconfigurations by non-technical personnel.

In order to fulfill these needs, an architecture was designed composed by a Wireless Sensor Network (WSN) [22] and a Computational Platform (CP) [23] accessible via the Internet. The WSN comprises a set of instrumented mobile nodes carried by sheep (collars); a set of statically placed nodes for defining the grazing area and collect sheep’s data (beacons); and a gateway that, besides connecting the WSN to the CP, delivers local services. Due to the application constraints, all WSN components were customized, avoiding the use of IP-based solutions up to the gateway level. The gateway was also developed specifically for this application, allowing a smooth integration between the local WSN and the computational platform (IP-based), as well as the integration of a local management service, to enable users to easily interact with the system for management and supervision procedures.

In order to support other intelligent farming solutions with similar requirements of SheepIT, a new M2M protocol stack was proposed by Temprilho et al. [24]. This paper extends that work significantly, by improving the state of the art review and presenting a more complete and thorough discussion of the proposed stack. Furthermore, a special focus is given to gateway, detailing its design, implementation and validation.

The remainder of this paper is organized as follows. Section 2 surveys the related work and analyzes the suitability of the existing protocol stacks in the context of SheepIT. Then, in Section 3, we present the network stack architecture and discuss the differences with respect to the state-of-the-art approaches. A special focus is given to the messages exchanged on the local network domain and to the gateway design. The feasibility of the approach is evaluated in Section 4, particularly in what concerns the gateway scalability. Finally, Section 5 presents the conclusions.

## 2. Related Work

The IoT domain embraces a huge number of applications with different goals, features, requirements and constraints. Consequently, to solve the specific needs of such applications, a wide range of communication models, protocols and standards have been emerging, being either applied on WSN, M2M or Cyber-Physical Systems (CPS) [7]. Considering the needs of the agricultural sector, the first part of this section presents a review of the most relevant communication technologies, protocol stacks and standards. Additionally, we evaluate if their use could meet the SheepIT project needs, as well as other related intelligent farming applications. Then, we perform a survey on existing animal monitoring solutions for livestock support, focusing on their distinctive features.

### 2.1. IoT/M2M Protocols and Communication Technologies

IoT encompasses an ample number of communication networks, either WSN, M2M, CPS, or a combination of them. From these, M2M communications are seen as the foundation of IoT platforms [6] since they allow the ubiquitous and autonomous communications between devices without human intervention [25].

M2M networks comprise both wired and wireless solutions but, the increasing need of mobility and scalability, together with the inherent high installation costs of wired solutions, have led to an increasing number of M2M wireless options. These can be organized in three groups (Figure 1): firstly, the *capillary* solutions [26], where a local network coexists with a local gateway that makes the interaction with Wide Area Networks. Some examples are the Wireless Local Area Networks (WLANs) such as WiFI or HiperLan, and the Short-range Wireless solutions such as Bluetooth, ZigBee and Z-Wave. These solutions are characterized by a data rate that can go from bits per second to megabits per second; secondly, the popular cellular networks, which have a greater communication range compared to the capillary solutions, being the 2G, 3G, 4G or more recently the Long Term Evolution (LTE), some examples; finally, addressing the specific IoT/M2M communication requirements, particularly power consumption and autonomy, a new set of technologies started to gain relevance, the denominated Low-Power Wide-Area Networks (LPWANs). These technologies combine the wide range capability of cellular networks with the lower power consumption of *capillary* solutions [27].

Cellular networks and the WLAN’s were not considered to be used in the scope of the SheepIT project, due to their higher power consumption and price. Thus, our targets are the short-range wireless communications and LPWAN solutions. Among these, some may resort to gateways to interact with other networks while others resort to proprietary protocols.

This heterogeneity of approaches has led to the definition of distinctive IoT protocol stack models [7] with three, four or five layers. The simplest one is based on three layers: (i) The Perception Layer, which is the lowest layer and includes all the functionalities related with physical sensors; (ii) The Network Layer, which is responsible for ensuring a seamless transmission of the data gathered on the Perception Layer to the Application Layer; (iii) and the Application Layer that allows the customization of different services according to user requirements and needs. However, some authors consider this approach too simplistic and hence a model of five layers gained popularity [8]. On this model, besides the Perception and Application Layers, already included in the three-layer model, three other layers are included, namely: (i) Transport or Object abstraction Layer, responsible for receiving the data gathered on the Perception Layer and making it available to upper layers; (ii) Processing, Service management or Middleware Layer that deals with the huge amount of data that is gathered and with the big heterogeneity of objects within the Perception Layer; and (iii) Business Layer, that allows the management of the whole IoT system.

Notwithstanding, due to the interoperability and popularity of IP, many practical deployments of IoT applications follow a typical four layer TCP/IP architecture (Physical and MAC Layer, Network Layer, Transport Layer and Application Layer) with the necessary adaptations, particularly in terms of the protocols that can be used on each layer. Consequently, and as our proposed stack is also based on four layers, the following survey tackles the most popular protocols used on those layers, as suggested by Naik [9] and illustrated on Figure 2. Furthermore, after reviewing those protocols we also survey two additional approaches. On the one hand, we survey the LPWAN technologies because of their potentiality when applied to IoT/M2M scenarios. On the other hand, we review Non-IP based stacks, which are relevant because there are IoT application domains with energy, processing and radio bandwidth constraints that are so severe that they cannot be handled by the remaining methods. Finally, and since the main contribution of this paper is the development of an IoT Gateway capable of addressing the SheepIT project requirements, we also provide a survey on existing IoT Gateways and highlight the specificities of our solution.

#### 2.1.1. Physical and Mac Layers

IEEE 802.15.4 is still one of the most popular standards within the Physical and MAC layers. It was designed with three main requirements in mind: low consumption, low complexity and low cost [28]. Hence, it was adopted by several relevant protocols, such as ZigBee, ISA100 and WirelessHART. It operates mainly in the ISM band of 2.4 GHz, although it can also operate in the 868 MHz (in Europe) and 915 MHz (in the United States of America) bands. It supports data rates up to the 250 kbps and, concerning the MAC Layer, both Carrier Sense Multiple Access/Collision Avoidance (CSMA/CA) and Time Division Multiple Access (TDMA) (in the beacon-enabled mode) are supported. However, the literature reports significant limitations of the latter mode, particularly regarding the network formation [29] and mobility [30]. Besides IEEE 802.15.4, several alternatives emerged at the Physical Layer, but are not directly inter-operable with IP, requiring intermediate layers or Gateways. Examples are Z-Wave, designed particularly for domotics [31] and EPC-Global for RFID technologies [32].

#### 2.1.2. Network Layer

Regarding the Network Layer, the Internet Protocol (IP) is the predominant choice, with two coexisting versions, IPv4 and IPv6. The latter one was born due to the exponential increase of the number of addressable devices and consequent exhaustion of available IP addresses. Consequently, the introduction of IPv6 was inevitable, being introduced gradually on all networking devices. Nevertheless, and despite its popularity and interoperability, its direct use on IoT devices is not always reasonable. The overhead of such protocols (and inherent processing needs), usually clashes with the constraints associated to IoT devices. Thus, a big effort was made by the 6LoWPAN working group (WG) from IETF, in order to minimize the IPv6 limitations and to make it suitable to be used by IoT devices. As a result of such work, the 6LoWPAN protocol was released [11], defining the specifications of the Network Layer of IP-based WPANs based on IEEE 802.15.4 physical and MAC layers. This protocol defines an intermediate layer between the MAC and Network layers, the Adaptation Layer (AL), that aims at ensuring the interoperability between the referred layers, implementing compression, fragmentation and reassembly mechanisms. Despite the advances attained by this AL, the complexity introduced for supporting IPv6 is still too high for technologies that excel by their simplicity, low power consumption and low overhead. Some examples are the Bluetooth Low Energy (BLE) and Near Field Communication (NFC). Hence, the 6lo working group extended the 6LoWPAN protocol to these two technologies, establishing the IPv6-over-foo AL [33]. Pursuing similar goals, Farris et al. [33] proposed a similar solution for integrating RFID systems into IPv6 networks. Notwithstanding, and despite all the efforts, there will always exist a significant overhead associated to the implementation of such ALs, which may make them inappropriate for tightly constrained applications.

Still within the Network Layer, the Routing Protocol for Low Power and Lossy Networks (RPL) [34] is the main option considered when deploying routing on 6LoWPAN solutions. This protocol was specified by the IETF Routing Over Low Power and Lossy Networks WG (ROLL) and it briefly consists on a distance vector routing protocol that uses Destination Oriented Directed Acyclic Graphs (DODAG) to define routes. The construction of the graph is made through an objective function (OF) that dynamically defines the routing metrics to be used taking into account the network constraints. In addition to the support of different traffic flows, as point-to-point, point-to-multipoint and multipoint-to-point, RPL adapts itself to the network rate and accepts routing metrics such as link quality or current battery status of devices, in exchange for a higher computational cost.

#### 2.1.3. Transport Layer

The Transport Layer, also known as Host-to-Host Transport Layer, is directly transposed from IP to the IoT domain. Here, the two most important protocols are the Transmission Control Protocol (TCP) and the User Datagram Protocol (UDP). TCP is a connection oriented protocol that ensures a reliable data delivery with end-to-end error detection and correction in exchange for a higher overhead and lower data transmission speed compared to UDP. Contrary, UDP is a connection-less low-overhead protocol that does not ensure any kind of acknowledgement, privileging the data transmission speed over the data transmission quality. Both protocols are commonly used by different applications and its choice is closely dependent on the application requirements.

#### 2.1.4. Application Layer

The Application Layer creates the final level of abstraction that allows the development of different kinds of user applications. In addition, the strict computational and energy constraints of IoT devices prompted the emergence of several lightweight application layer protocols, such as CoAP, AMQP, MQTT and HTTP, to cite just a few.

COAP [12] is based on a request/response architecture and runs over UDP. As it is based on HTTP methods, it is interoperable with HTTP. For security, it uses the Datagram Transport Layer Security, similar to TLS in TCP. AMQP [35] is based on an asynchronous publish/subscribe architecture, runs over TCP and uses TLS/SSL to ensure security. It provides Quality-Of-Service (QoS) guarantees, specifically: *at most once, at least once and exactly once*. As AMQP, MQTT [36] also runs over TCP, it is also based on an asynchronous publish/subscribe architecture; it also uses TLS/SSL to ensure security and it supports three types of QoS, in this case: *fire and forget, delivered at least once and delivered exactly once*. HTTP is a popular web messaging protocol based on a request/response architecture [37]. It runs over TCP, uses TLS/SSL for security and does not define any QoS. Contrary to the aforementioned protocols, HTTP does not define the header and payload sizes, being dependent on the web server implementation. The choice of the protocol to be used is a daunting task and depends greatly on the kind of application and devices used. Thus, normally, there is not just one possible correct choice [9].

#### 2.1.5. Low-Power Wide-Area Network Technologies

An emergent area on low power communications respects the LPWAN technologies. They operate in the license-free bands of the spectrum and can be positioned between the cellular networks and the short-range communications, taking the best of each one. They potentially cover communication ranges similar to a cellular cell, keeping a low power consumption like some of the short-range communications technologies. The data rates are also similar to the short-range communication solutions and, despite being smaller than WLANs and cellular networks data rates, are adequate for many IoT solutions. Some examples are Sigfox, Long Range (LoRa) WAN, NarrowBand-IoT (NB-IoT) and LTE-M.

Sigfox [16] is a centralized, cellular-like, low-throughput wireless communication system. Energy-wise, it is very efficient, but it provides a limited capacity (a few messages per day with up to 12 bytes each). It requires cellular coverage which, at least in Portugal, is based on the deployment of gateways on existing public telecommunications base-stations. Its actual coverage is still relatively small, particularly in rural areas. It does not allow to perform localization and the bandwidth is insufficient for reporting the amount of data required by the SheepIT project.

LoRa [19] is a Wireless Wide Area Network (WWAN) specification designed to allow long range communications for IoT-like applications. It was designed to minimize energy consumption and provides modest bit-rates (300 bps to 50 kbps per channel). It may be used as a private network, but it may also be used to deploy a shared service infrastructure. LoRa provides a reasonable bandwidth and is energy-efficient. It also permits RSSI-based localization, but its performance in this regard is poor. For example, a practical evaluation in a limited area (110 × 64 m) and with an infrastructure based on six anchor nodes, impracticable for farming applications, generates errors of several meters [38]. LoRa also has a Time-of-Flight based localization scheme that is proprietary, closed source and can only be used in a black-box manner. Results on its accuracy are scarce, but indicate a poor performance, at least for wide ranges (e.g., 100 m for stationary nodes using a public location service [39]). Moreover, LoRa prescribes a star topology and constrains the kind of synchronization and communication between nodes and the gateways (e.g., for battery-powered devices—class A uses an ALOHA-like protocol, without synchronization, and with dedicated receiving slots that may not be needed). Finally, LoRa nodes were considered too expensive to be implemented in each of the animal sensors.

NB-IOT [17] and LTE-M [18], part of 3GPP Long-Term Evolution (LTE), have been developed for M2M and IoT applications. Despite providing relatively high bandwidth and potentially low consumption, these protocols depend on a public communication infrastructure, which is often unavailable in rural areas, so it is unsuited for the objectives of SheepIT. Moreover, the need for paying a fee to the telecom operator by each animal eventually may lead to unbearable exploration costs.

#### 2.1.6. Non-IP Protocol Stacks for M2M

There are several applications that avoid the use of IP-based solutions in order to optimize resources, processes and communications. Therefore, there are several approaches that resort to non-IP communication solutions, either resorting to ALs, implementing communication stacks as a whole, or both.

ZigBee is a practical example of a full communication stack. It is based on the IEEE 802.15.4 standard for the Physical and MAC layers, the higher layers being defined by the Zigbee Alliance [19]. It aimed at offering a low power wireless communication mechanism and gained particular popularity on WSN deployments. However, as introduced in Section 2.1.1 IEEE 802.15.4 presents relevant constraints when considered for TDMA scenarios.

BLE is another popular and trendy solution for low power communication solutions. It is an enhanced version of the classic Bluetooth, allowing communication ranges 10 times higher (around 100 m), lower radio power and lower latencies. Its use is more popular on WSN applications [40] and vehicle-to-vehicle communications [41], but besides its effective reduction of power consumption and increased range, it still presents severe limitations on both star and mesh topologies deployments [42].

EPCglobal and Z-Wave can both be used as fully integrated solutions or by the mean of adaptation layers (as introduced in Section 2.1). The former is relevant for UHF RFID devices while the latter was designed for home automation applications.

In sum, there are a multitude of solutions to deploying an IoT/M2M application, either in terms of protocol stacks or communications technologies. The standardization efforts regarding communications on those domains are impaired by the high heterogeneity of applications and their requirements.

#### 2.1.7. IoT Gateways

The diversity of IoT applications, devices and protocols, creates a natural lack of interoperability. Consequently, to allow the communication of devices that rely on different communication technologies or devices that use different application protocols, an intermediate device is required, typically known as an IoT gateway. In addition, several solutions can be found in the literature, some based on academic research projects that aim at solving particular problems, while others consider bigger research and standardization proposals.

In [43], an IoT gateway that enables the interoperability between Zigbee and GPRS protocols is presented, being tailored for applications composed of light, temperature and humidity sensors. A similar work is proposed by [44], with an extended number of supported devices and protocols. This Smart IoT gateway allows a multifunctional configuration and supports protocols such as Zigbee, RFID and RS485. Furthermore, considering the popularity of smartphones, a smartphone gateway is proposed in [45]. It supports several protocols like Wi-Fi, ANT+, Bluetooth, NFC, ZigBee and 6LoWPAN. A similar approach is followed on [46], being presented a smartphone-centric solution on which the main links supported on the connection to sensors are based on BLE and IPv6. Furthermore, on [47], a wireless IoT gateway is proposed, mainly supported on APIs composed of Restful [48] web services. The gateway architecture, besides the REST API, includes dynamic device discovery in order to allow the insertion or removal of devices and a management connection module to handle devices that do not support REST.

Besides these academic research works, several standards that specify the deployment of IoT gateways for supporting M2M/IoT services have been developed. One of the most relevant ones is the OneM2M [49], that has been developed with the participation of important standard development organizations and consortiums (including, for instance, ETSI, TIA and Broadband forum), and powerful companies (e.g., Cisco, Intel, Samsung, and Ericsson). OneM2M provides a horizontal platform that supports secure, reliable and efficient [20] operations of multiple IoT/M2M services, particularly those relying on REST APIs. It supports multiple existing application protocols such as HTTP, CoAP and MQTT, and communication technologies such as ZigBee, Bluetooth, WiFi and Cellular networks on the sensing domain. To turn agile and normalize the management operations, OneM2M also supports the OMA-DM and BBF TR-069 protocols. Other turnkey solutions are available, being differentiated by the application protocols and communications technologies that they support. Intel [50] provides a gateway that supports protocols such as MQTT, WiFi, VPN, Bluetooth, Cellular technologies and Zigbee, but also Serial and USB interfaces. SmartM2M [51] supports HTTP, COAP, Cellular technologies, Zigbee, Bluetooth and Wifi. The Lightweight M2M (LWM2M) [52] supports COAP and 6LowPAN technologies.

In sum, we can find a wide range of IoT gateways, some specially designed for certain applications, while others try to offer an horizontal integration of a diversity of IoT/M2M protocols and communication technologies. In common, they share the goal of creating an integrated IoT platform that could support multiple applications, protocols and standards. However, none of them fulfill completely the requirements of the SheepIT project, mainly in what concerns the local support of application-level services, such as an improved localization mechanism and system management. Even if some of those solutions could be customized, the cost of such integration could be economically unaffordable. Furthermore, and despite the great advantages that a fully integrated IoT/M2M gateway could offer, the resources that it takes can be superfluous considering the intelligent farming scenario, where the simplicity, low cost and high autonomy are the key features.

### 2.2. Animal Monitoring Platforms

The monitorization of animals, namely sheep, is one of the main goals of SheepIT. There are several solutions available for different animal breeds, both in the literature and commercial domains. Typically, these solutions are split into two main groups: location monitoring and behaviour/activity monitoring. The former one allows farmers to keep track of animals and thus infer preferred pasturing areas, grazing times or even detect absent animals. The latter group focuses on detecting the type and duration of an animal’s activities, such as resting, eating or running. Albeit both kinds of monitoring tasks are relevant for the context of SheepIT, activity monitoring is particularly important since it is the basis of the conditioning mechanism that it implements.

The monitorization of animals’ locations was one of the first technologies implemented in the livestock sector. The unpopularity of shepherds’ profession, together with the demanding livestock productivity, has led to the deployment of automated solutions for monitoring the grazing areas and duration. Consequently, several solutions arose to address such issue, being the Global Navigation Satellite Systems (GNSS), especially the Global Positioning System (GPS), the most popular technology used. Some examples are the Australian OzTrack platform that allows to upload GPS gathered data (evaluated with on crocodiles) [53], the e-Pasto platform that offers an application to monitor the location of cattle grazing on mountains [54] and the Digitanimal [55], a solution that offers a vast inventory of GPS-based devices for monitoring the location of different animal species (e.g., cattle, goats, sheep, dogs, etc.). These solutions share the use of GPS and resort on web platforms and/or mobile apps to interact with the user. However, they only perform monitorization, and thus still require the use of expensive physical fences. To suppress such a need, some solutions proposed the addition of a virtual mechanism to the existent localization solutions. Two relevant examples are the eShepherd solution [56] applied to grazing cows, and the NoFence platform, developed specially for goats [57]. This later one, available commercially, was tested both on sheep [58] and lambs [59], and the conclusions point to the existence of technical issues and several difficulties on teaching sheep to stay inside the virtual fence. Thus, there are still improvements that must be made in this area.

Despite the evolution of GPS technology and developments on energy harvesting, duty-cycle and sensor fusion techniques, to reduce the power consumption and increase device’s autonomy, the impact of GPS location is still very relevant. One alternative that we consider interesting for animal monitoring, is the use of RSSI. As it is present on most Radio Frequency (RF) technologies, its deployment would reduce energy consumption and contribute to the reduction of the cost and size of the hardware, which are key factors. Such use is reported on Nadimi et al. [60] and Huircán et al. [61], when applied to cattle location monitoring. Despite being an attractive solution, these works revealed a still significant immaturity of this kind of solutions, particularly concerning the lack of precision. Thus, there is still an open research opportunity in this area. Despite not being addressed in the scope of this paper, it is a goal of the SheepIT project, and thus, it was taken into account in the design of the solution.

Concerning the behaviour and activity monitoring, the most usual automated approaches are based on accelerometry and audiometry. Tri-axial accelerometers allow measurement of both static and dynamic accelerations, which, after being combined, processed and analyzed, permit to distinguish different kinds of animal behaviours and activities. Furthermore, these data can be stored and processed through advanced and powerful processing mechanisms, as Machine Learning (ML) algorithms, allowing the extraction of additional knowledge, not directly available on raw’s sensor data.

A few practical examples are the study of activity patterns of Griffon Vultures [62], the classification of goats activity [63], the classification of cattle activity [64,65,66] and the classification of sheep behaviour [67,68,69]. Despite a few adaptations, necessary to accommodate the specificities of the application scenarios (i.e., the animal breed whose behaviour is to be studied), all the works rely on a similar approach, namely on gathering field data and transmitting it to computational platforms with advanced processing capabilities. The existing solutions differ mainly on the mechanisms they use to gather the data (through WSNs, external memories, cellular technologies) and on the algorithms they use to process the data (Machine Learning classifiers, Artificial Neural Networks). SheepIT requires that the behaviour detection must be carried out in Real-Time, and thus must be deployed at the box placed on the animals, therefore resulting in low-cost and low-power hardware, impairing the use of the advanced data processing mechanisms carried out by the aforementioned works.

The inherent advantages offered by animal monitoring activities to livestock productivity has led to the emergence of several commercial solutions. Some relevant examples are CowScout [70] and Cowlar [71], that aim at monitoring and quantifying the activity of cows in order to improve the lactation and insemination processes. However, these commercial solutions are scarce in details and it is only possible to state that they allow detecting animal activity, probably relying on accelerometers’ data.

In summary, there are several solutions that tackle some individual (or group of) issues that SheepIT aims to solve. Table 1 summarizes the most relevant solutions surveyed. As observed, none of the presented solutions is capable of detecting and conditioning sheep’s posture while grazing. An additional comparative study of some of the presented solutions is given by Nobrega et al. [72].

## 3. IoT-Based Stack Architecture for Intelligent Farming Solutions

The solution developed in the scope of the SheepIT project aims at, on the one hand, supporting the integration of a set of sensors into mobile nodes coupled to animals and, on the other hand, supporting the integration of a set of static sensors distributed along vineyards for monitoring plants and soil parameters.

The requirements imposed by the uncertainty of animals’ locations, their extremely gregarious behaviour and the effect of the terrain relief in the radio coverage, lead to the design of a new solution, from the IoT devices to be deployed, up to the IoT communication network. In this section we present the overall system architecture, focusing on devices’ operation and communication network.

Figure 3 illustrates the overall system architecture. Besides the novelty introduced by some elements (e.g., collar device carried by sheep), the system’s architecture follows a typical IoT architecture, being composed of two main modules. The first is a WSN that implements all the critical local functions of the system, such as the animal behaviour monitoring and conditioning. Additionally, it enables the sensor data gathering that is then forwarded to the user. The second module concerns the CP, hosted for instance on the cloud, and that has as functions, message brokering, data analysis, data storage and data display, the latter through appropriate user interfaces. The following subsections tackle both modules, as well as devices and messages exchanged.

### 3.1. WSM and M2M Components

The WSN is composed of mobile nodes for animal monitoring and fixed nodes for farming sensing, for instance, temperature sensors, weather forecast stations, humidity sensors and light sensors. Animal sensors (e.g., neck inclination, distance to ground) are implemented on a collar coupled to the animals’ neck. Besides monitoring and condition the animals’ behaviour locally, the sensed data is reported to the network through a radio transceiver. Further details on collar’s operation and design are given in [74].

The fixed nodes, dubbed beacons, besides implementing the needed farming monitoring sensors (adjustable accordingly to the specific needs of the farm), act as infrastructural networking devices. Consequently, they are responsible for collecting the data sent from sheep sensors and for relaying it through neighboring beacons in order to ensure its delivery to one or more aggregating points (gateways). Additionally, they ensure the network synchronization and their periodic communications are designed to allow the establishment of a RSSI-based localization mechanism [74].

The M2M communications within our WSN are implemented by a protocol stack (illustrated in Figure 4) specifically designed to satisfy the SheepIT requirements [22,24]. Despite having this particular project in mind, the stack design was driven by more general goals, aggregating the traffic requirements of several applications related to farming management and supervision. Our approach follows a four-layer architecture (Physical Layer, MAC Layer, Transportation Layer and Application Layer) since it is the one that enables a straightforward match between the defined requirements and each one of the layers. Below, we outline the layers that compose the stack as well as their purpose.

Regarding the Physical Layer, the major constraints are associated to the wine-growing and similar lands, mainly in the Douro’s region (Portugal) where the terrain is very rugged, with a set of valleys and irregularities carved in the landscape. Such constraints stopped us from considering existing communication standards with Physical Layers based on higher frequencies (e.g., IEEE 802.11, BLE and some implementations of IEEE 802.15.4 and ZigBee). As, theoretically, lower frequencies (and thus higher wavelengths) present better propagation performance, our Physical Layer is based on radio transceivers operating preferably at the ISM band of 433 MHz. Even so, and due to the popularity of the 868 MHz band, we recently added this option.

The MAC Layer was an object of careful design due to its implication on the power consumption and consequent autonomy of networking devices, particularly in the case of mobile nodes (sheep collars). The strategy proposed approaches a Time-Triggered (TT) architecture, which means that relevant tasks and communications are executed following a predetermined scheduling. Hence, this scheduling may be defined aiming at optimizing the entire system’s operation. Regarding the wireless communications, considering that most of the communications are periodic in nature and in order to avoid collisions, packets losses and consequent waste of bandwidth and energy consumption with retransmissions, permanent-state communications are TDMA-based. This communication scheme is complemented with CSMA-based sporadic communications, used for events such as device’s registration on the network (after a system boot, after a reset or after a long period of non-reporting). To this end, a periodic-based structure is used, on which a sequence of micro-cycles (µC) form a Macro-Cycle (MC) that is repeated over time. Each type of µC has a particular purpose and, accordingly to that purpose, a TDMA/CSMA MAC policy. As depicted in Table 2, three types of µC are currently defined, albeit more can be easily added if required.

Independently of the type of µC considered, all respect a common structure that is represented in Figure 5. They start with a *Synchronization Window* (SW), during which each beacon sends a synchronization message carrying its identification and information about the current type of µC. This information is used by the remaining devices to synchronize with the network and allows the collection of RSSI information from multiple devices, crucial for the localization algorithm. Following the SW, there is a *Turn Around Window* (TAW) for processing purposes, namely for acquiring and processing sensor data and preparing subsequent tasks. Finally, the *Variable Traffic-type Window (VTW)* is where the actual communications do occur. More details about this communication scheme may be found in [22].

To enable the implementation of the aforementioned communication scheme, several messages were defined, composing our Message Transport (MT) module. Figure 6 illustrates the existing interactions between networking elements for C2B and B2B µC. As explained before, independently of the µC type, there exists a initial phase, the SW, where beacons broadcast *BeaconSynchro* (BS) messages. During this window, all devices are awake and, upon receiving these messages, they are all able to identify the starting and type of the new µC, triggering the necessary calculations in order to know if, when and what they should transmit. This allows constrained devices (i.e., collars) to enter in energy-saving modes right after finishing their tasks. After this initial phase, the exchanged messages depend on the type of the µC. For the C2B case, the remainder of the µC is devoted to the transmission of *Collar-to-Beacon* (C2B) messages, one per collar and per time slot. Only beacons listen to these messages, being the remaining collars in power-saving modes. On the other hand, if the µC is of type 3 (B2B), the messages exchanged are called *Beacon-to-Beacon* (B2B) and are also transmitted in a time multiplexing way, being assigned to each beacon one or more time slots, accordingly to the maximum number of beacons and to the routing scheme defined.

To enable a certain level of network dynamism, particularly the insertion and removal of devices while the system is running, a Network Manager (NM) module is included. Additionally, since some sensors are coupled to animals and thus movable on uneven terrain, it is of crucial importance that the gateway monitors sensor communications and detects missing animals or malfunction devices, such that they can be excluded from the network. Figure 7 exemplifies the exchanged messages when a new collar needs to be registered on the network’s system. As these events are sporadic, a specific CSMA-based µC (PR), is reserved for this kind of events. The collar initializes a new registration procedure by sending a *Registration Request* with information about the serial number of the collar and the identification of the animal (i.e., animal identification information present in their stomach bulb transponder [76]). These requests are propagated through the entire network until reaching the gateway. The gateway validates the request and, if possible, sends a *Registration Reply* with the network identification and the serial number that is propagated backwards to the collar.

The Routing Manager establishes the active routing scheme (within the beacon infrastructure). Beacons act as routers and they establish adjacent relations with their neighbors, based on the signal intensity monitoring. To simplify the implementation, currently, the protocol just considers one type of routing metrics, namely the distance between devices (shorter distances, smaller routing cost).

The Security Manager (SM) needs to ensure that only the data sent by allowed devices is accepted. To do so, besides the sensor registration scheme described in the scope of the NM module, a specific code, associated with the property where the devices are being used, is incorporated on all messages. This value is used by devices to know if a certain message can be accepted or not. The registration scheme allows the infrastructure to authorize new sensors and request corresponding schedules.

The Application Layer implements several farm monitoring applications, mainly envisaged for vineyard and orchards scenarios. It includes animal sensing (SheepIT is the starting point example), plants and weather monitoring, as well as the configuration of several operational details of the application. This diversity of applications allows to offer an integrated monitoring platform for intelligent farming applications, supporting the management and decision making activities within a farm.

### 3.2. Gateway

As depicted in Figure 3, the interface between the WSN, the CP and the User Interface domains is carried out by the gateway. In addition to the standard gateway functionality, which is essentially joining networks to allow devices to intercommunicate irrespectively of the network where they belong, the SheepIT gateway supports additional services developed to address the specific requirements of farming applications, as referred to in Section 2.1.7.

Therefore, the most elemental functionality of the gateway is forwarding data between the WSN, the CP and the User Interface. This activity requires minimal processing, consisting essentially in decoding the data frames coming from each one of the interfaces, placing the data in a local database and triggering the processes responsible for fetching the data and sending it to the target interface(s).

Many farms are located in remote places and, as such, permanent access to the Internet cannot be considered as granted. Moreover, there are some services that require low response times and, as such, it is essential to deploy them as close to the data sources as possible. Consequently, the gateway should support locally at least a few critical services, namely:**Alarms:** alarms should be raised as quickly as possible in the event of anomalous situations. The gateway collects data from all the animals, so it has a global view that allows it to correlate the received data, thus being a privileged place to detect abnormal situations. For example, if all animals suddenly exhibit a running behavior that may indicate that the flock may be under attack of a predator, and thus an alarm should be raised. A sheep that disappears from all the beacons can have escaped, and so an alarm should also be raised;**Local management:** local web interface, to allow performing on-site administration tasks (e.g., defining the height at which animals are allowed to feed, if the restrictions are active or not, etc.), consult real-time and historical state data, among others. These operations shall be possible through a WiFi or Bluetooth enabled PDA or smartphone, without requiring access to the CP’s web interface;**Technical management:** local web interface, to enable technical interventions, such as tuning the communication parameters to the specific characteristics of the exploration (kind of terrain, dimension of the flock, etc.), as well as to allow system-level debug via the network.

The number of gateway devices is expected to be small (just one, in most of the cases). Moreover, they are not moved frequently and are usually deployed in accessible places. Therefore, we assume that gateways are not subject to the stringent power, size and energy consumption limitations that beacons, and particularly collars, face. Thus, gateways can use more powerful hardware, both in terms of processing and storage capacity, which is essential to enable the realization of the services mentioned above. Considering the complexity of the services to be supported, we based our proposal on an Embedded PC platform running Linux. As illustrated in Figure 8, the gateway comprises Internet and WiFi/Bluetooth, made available through suitable network interface cards. The Gateway also contains a beacon module, interconnected with the Embedded PC via a serial interface. The beacon module ensures compliance with the communication scheme of the WSN. Combined, these interfaces abstract the communication services offered to the local services.

Figure 9a illustrates the gateway’s WLAN interface processing flow. After system initialization, which includes initialization of data structures and hardware, the system waits for the arrival of frames through the serial port. When a new frame arrives it is subject to pre-processing, which includes validation, parsing and time-stamping. The processed data is then stored on a local database. Afterwards, the processing flow forks into two different lines. The left side one corresponds to the Localization procedure. To improve the accuracy and stability, the localization procedure filters and combines the RSSI data from both slaves and beacons [74]. The improved animal’s estimated position is then stored on the local database. The right side flow corresponds to the alarm generation. The incoming sensor data is analyzed to detect abnormal behaviors. Relevant data includes localization, posture and activity type, penalization counters, inertial sensors readings, among others. A historical record of this data is maintained for each one of the animals, and the algorithms can access both individual and global data. Different algorithms can be applied, according to the specific use-case requirements, but common alarms include panic situations and missing animals. Eventual alarm events are stored at the local database. When the incoming data is completely processed, the relevant fields (sensors, timestamp, error counters, alarms, etc.) are fetched from the database and sent to the CP. The CP is designed to receive data using the AMQP protocol, thus, the gateway implements this protocol at the Internet interface. Further details on the CP operation are given in Section 3.3.

Figure 9b illustrates the operation of the technical and management web interfaces. After system initialization, the gateway awaits for the arrival of a frame from the WiFi/Bluetooth interface. When a frame arrives, it is validated and parsed. Frames can be of one of two classes, *GetData* and *SetData*. *GetData* frames are sent by the user device and correspond to data requests. Conversely, SetData frames correspond to parameter update commands. When a *GetData* frame is received, the Gateway fetches the corresponding data from the internal database and reports it via the same interface. *SetData* frames correspond to modifications to the system operation (e.g., activate or deactivate the posture control, set the height below which animals are allowed to feed, define the radio transmission power, etc.). When one of these frames is received, the corresponding parameters are stored on the local database and then forwarded to the WSN, thus reaching beacons and collars.

### 3.3. Computational Platform (CP)

The CP (Figure 10) receives, processes and stores the data gathered by facilities’ gateways and makes their information available in a summarized way to the human operator. In order to guarantee scalability with the growth of the number of facilities, animals and equipment, it was designed using an AMQP broker that allows the subscription of message topics received by several entities of the solution. The topics distributed by the RabbitMQ-based broker are subscribed by a data-processing engine, the Apache Spark [77], that persists real-time data stream into the platform’s data repository. Moreover, such infrastructure allows the incorporation of data mining and machine learning libraries [23], which can be used to predict specific and relevant events such as potential diseases on animals or vines/orchards. Such data processing tasks are implemented as background tasks, accessing the repository using the previously persisted data.

The processing framework was enabled with a rule management server (Drools [78]) that also subscribes broker stream topics in order to implement a central alarm system that can be used complementary to the alarm generator module implemented on the gateway.

Finally, the CP also includes a RESTful API in order to allow a smooth deployment of web and mobile user interfaces. Additional details on CP’s implementation can be found in [23].

## 4. Experimental Evaluation

To evaluate the feasibility of the proposed solution, we need to demonstrate that our approach complies with all identified requirements. In [22] the requirements and design of the communication mechanism, especially regarding the Physical and MAC Layers were addressed and theoretically evaluated. This evaluation included the validation of timing and energy consumption constraints, being demonstrated by the theoretical feasibility of our approach. Later on, in the work of Temprilho et al. [24], the solution was transposed for a full M2M communication stack, although the focus was the practical implementation of the Physical and MAC Layers. Furthermore, this implementation was tested and validated under controlled conditions (i.e., on laboratory). Finally, in [23], a particular use case of the presented mechanism was used to collect real data such that we could resort to ML algorithms as a tool for achieving a suitable and efficient control posture mechanism. This use case required a specific configuration of the network in order to allow gathering data with higher rates than the ones required during normal system operation. Albeit further tests shall be performed in order to assess the feasibility of the proposed solution under real conditions, the results of these first works are promising.

As the successful deployment of the solution also depends on the gateway deployment, this paper focuses on this issue. Thus, the following results are a complement and extension to the previous works. Explicitly, we are interested in assessing if the design and implementation of the gateway are suitable and capable of supporting the processing and timing needs of the application, particularly when the number of devices increases.

### 4.1. Evaluation Scenario Conditions

We are interested in evaluating the behavior of the gateway under normal system operation conditions. Such conditions were defined in previous works considering the requirements and the design of the WSN [22], and the temporal conditions defined and explained in [24]. As these decisions are directly related with animal husbandry and livestock conditions, they were supported by partners of the project, namely the Agrarian School Of Viseu and Ramos Pinto, Lda. In summary, those conditions are:The µC duration is set to 6 s (due to localization needs) and the MC duration to 60 s (the needed data report periodicity);The arrangement of one MC is defined as follows: one µC type 1 (PR), one µC type 2 (C2B) and the remaining ones are reserved for µC type 3 (B2B);The area to be covered is set to 10 ha, since, according to the practical experience of our partners, it is the average vineyard size of potential users of the solution;The number of beacons needed to cover such area is 20;The max number of collars in one property is fixed to 1000;The serial port baudrate is set to 115,200;The gateway is connected to the Internet through an Ethernet connection.

To implement the gateway an Orange PI Zero (Shenzhen Xunlong Software CO (Guangdong, China)) [79] was chosen, due to its low cost and low energy consumption. This device features an H2 Quad-core Cortex-A7 H.265/HEVC 1080p, a 512MB DDR3 SDRAM, runs a Linux-based Armbian operating system and it costs around 10€. Furthermore, the gateway application was developed using the “C” programming language and relies on a multi-threading architecture.

Furthermore, as at the present stage of the project it is unpractical to perform the necessary tests using the maximum number of collars defined for a single property (1000), we implemented a generator of traffic that runs over a Linux platform and sends periodically (every MC), the expected data that would be generated by a defined number of collars and 20 beacons. That allows to easily evaluate the behaviour of the gateway under different traffic conditions.

In order to ensure that the gateway is able to receive, process and publish all the data without congestion, and consequently without losing data, we measured the duration of all tasks, from the instant that the data sent by a beacon is received by the gateway via the serial port, until the moment that it is published into the broker. Those tasks are: (i) the time needed to receive all the data through the serial port interface; (ii) the processing time of the data received, including the time needed to store it in a shared memory and the time spent by the localization algorithm; and (iii) the duration of the task responsible for generating the messages (in JSON format) to be sent over the AMQP protocol and for publishing them into the broker. It shall be noted that when publishing messages through the AMQP library installed on the gateway, the measured time includes the time needed for generating a message, the propagation time of those messages through the Internet until reaching the broker, and the propagation of an acknowledge message sent by the broker after receiving the message.

Albeit the worst case scenario is expected to occur with the maximum number of collars (1000), in order to understand the impact of increasing the number of collars, we relied on the implemented traffic generator to evaluate how those times evolve considering different number of collars (100, 200, 500 and 1000). For each one of these, we performed experiments that lasted for 300 MCs, thus around 5 h.

Table 3 summarizes the medians of the durations measured. For improving readability, we split data into Beacons and Collars. Besides the duration of the tasks previously mentioned, we also present the cumulative time spent by all tasks. As it can be observed, the time spent by the serial port to receive the data is the most restricting task with the conditions defined, and naturally, it increases linearly with the increasing number of collars. The same happens with the collars processing task as well as with the task responsible for publishing the data into the broker.

However, as can be seen in Figure 11, there are fluctuations with an important impact on the cumulative duration of the processing tasks. Thus, it is important to evaluate the worst case scenario, i.e., consider the maximum durations measured for each task. Table 4 summarizes those measurements. As can be seen, even considering the worst case measurements, the cumulative duration of all tasks evaluated is smaller than 6 s, thus well quite below the duration of an MC.

### 4.2. Discussion

As tackled in the previous section, the implemented gateway does not cause relevant traffic congestion considering the evaluation scenario defined. However, there are a few aspects that shall be analyzed in future works, namely:The tests performed considered an Ethernet connection between the gateway and the CP. However, most of real scenarios shall be based in cellular connections (2G, 3G, 4G). Since some of these kinds of connections present bandwidth and latency constraints, we are expecting higher values on the time needed to upload the data to the broker. Thus, a deeper study shall be performed in order to assess the minimum bandwidth necessary to avoid congestion on the gateway;Receiving data from the serial port is the most predominant delay source, particularly when the number of devices increases. Possible solutions include increasing the baudrate or use other interfaces (e.g., SPI);This work did not consider the impact of additional modules that interact with the local database. For instance, the local alarm generator and the local web socket modules may increment the processing overhead. If it turns out to be a problem, it is possible to update the Gateway’s computing microprocessor, without a significant impact in terms of price and energy consumption;The fluctuations observed may be a result of other concurrent tasks of the operating system. Consequently, future work shall consider the adoption of the real-time services provided by Linux;Evaluating the feasibility of the gateway’s implementation is a step forward in the evaluation of the entire IoT-based stack proposed for intelligent farming solutions. Notwithstanding, there is still work to be done on such stack. Particularly, the network, routing and security modules shall be further developed and evaluated;There is also a critical requirement that still needs to be evaluated, particularly the collar’s autonomy. On [22], a theoretical evaluation of such autonomy was presented considering a pre-prototype version of the solution, being expected 3224 h of autonomy with a battery of 2600 mAh. However, we believe that this may decrease under real conditions.

## 5. Conclusions

The IoT concept arose as a consequence of the exponentially increasing number of devices that are connected to the Internet. Nowadays, we can find IoT devices and applications in almost all sectors of the society. This wide range of applications, with distinct requirements and constraints, has led to the development of a plethora of protocols, standards and communication stacks, each one trying to solve specific requirements.

The popularity of the Internet rendered IP-based protocols to be the most popular ones. However, despite the efforts made to reduce the impact of the overhead associated with those solutions, as it happens with the 6LoWPAN AL, such solutions are still unsuitable for several energy constrained IoT applications. A practical example is addressed in the scope of the SheepIT project.

Using SheepIT’s requirements as a starting point, an IoT-based stack for intelligent farming solutions is proposed, from the Physical to the Application Layer levels. The overall system architecture includes a WSN, with a set of mobiles nodes, a set of static nodes, a gateway and a computational platform. Previous works addressed essentially the WSN domain, namely devices and communication mechanisms. In the scope of this paper, the gateway node was presented, being the functional modules highlighted. Furthermore, to evaluate the feasibility of its implementation, the duration of their processing tasks was evaluated for a number of different devices’ data. The results show that the gateway is able to transmit all the data gathered within the duration of an MC, even considering the worst case measurements and the use of low-cost and low-power hardware.

Future work shall include a similar evaluation on a 2G, 3G and 4G connection between the gateway and the Internet. We expect a significant increase in the latency when publishing the data into the broker, particularly due to the lower bandwidth of such a solution as well as its lower reliability. In addition, the impact of the local services shall be evaluated.

Additionally, and regarding the proposed stack, a more detailed analysis shall be performed to the modules of the Transportation Layer not addressed in the scope of this paper. Specifically, the routing manager and the security manager modules shall be detailed and evaluated. Finally, all the M2M components shall be subject to intensive experiments under real deploying conditions.

## Figures and Tables

**Figure 1 sensors-19-00603-f001:**
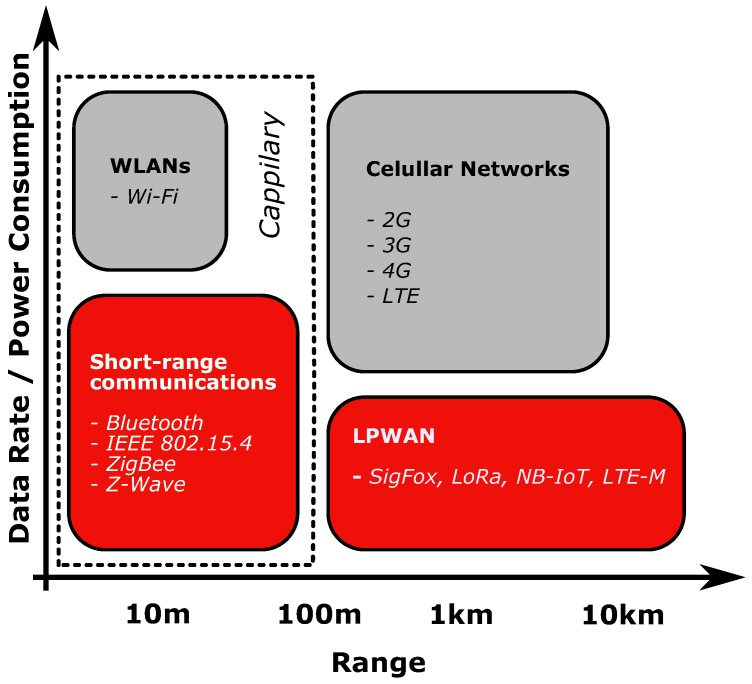
Communications in Internet of Things—major groups (based on [27]).

**Figure 2 sensors-19-00603-f002:**
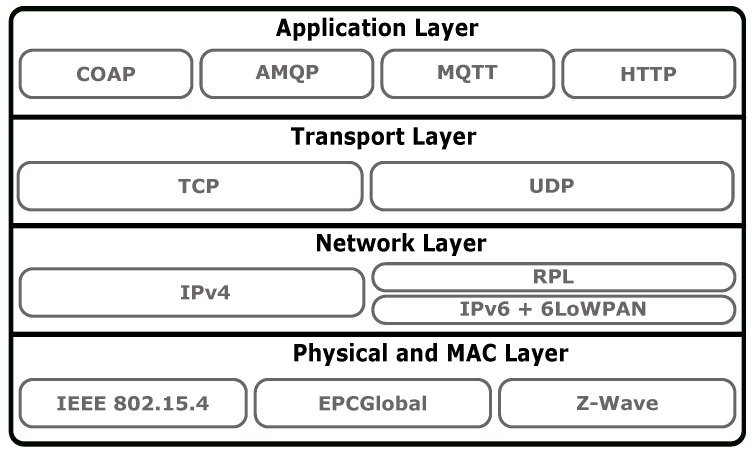
Popular IoT standards on an Internet Protocol (IP)-based solution (based on [7]).

**Figure 3 sensors-19-00603-f003:**
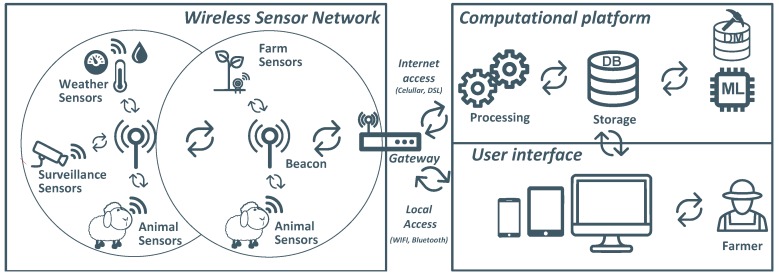
Overall system architecture.

**Figure 4 sensors-19-00603-f004:**
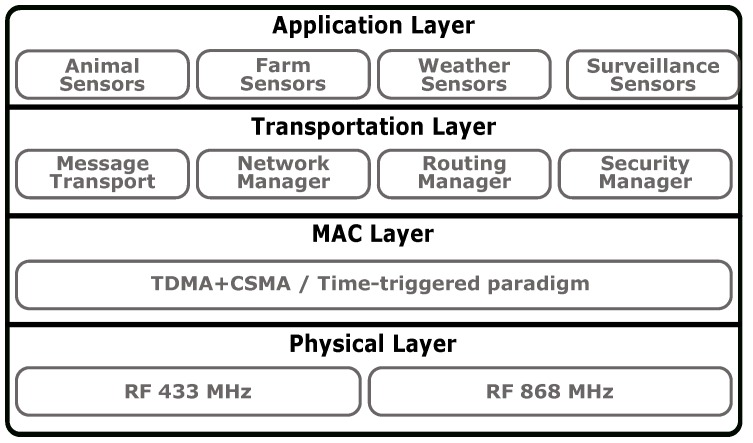
Intelligent farming Machine-to-Machine (M2M) Stack.

**Figure 5 sensors-19-00603-f005:**
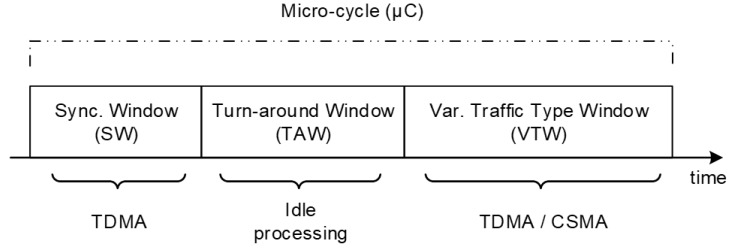
Micro-cycles (µC) structure [75].

**Figure 6 sensors-19-00603-f006:**
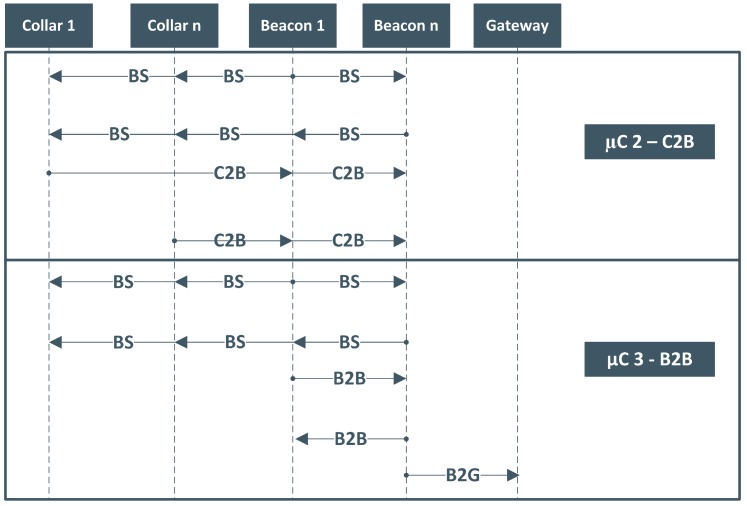
Message sequence chart of a µC type 2 and 3.

**Figure 7 sensors-19-00603-f007:**
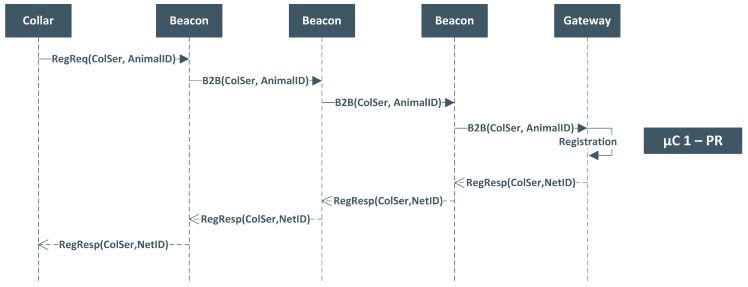
Collar Registration Message Sequence Chart.

**Figure 8 sensors-19-00603-f008:**
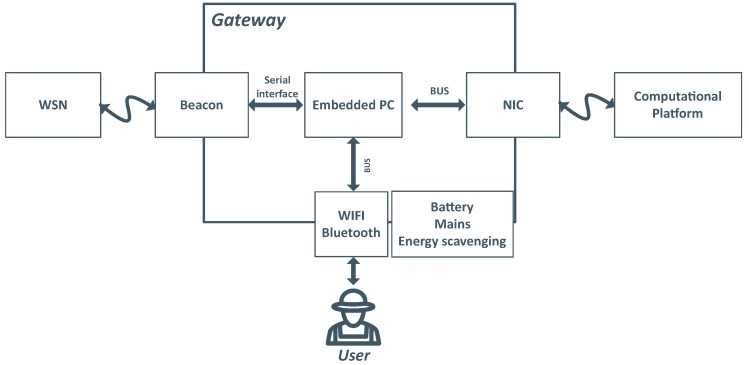
Gateway internal modules.

**Figure 9 sensors-19-00603-f009:**
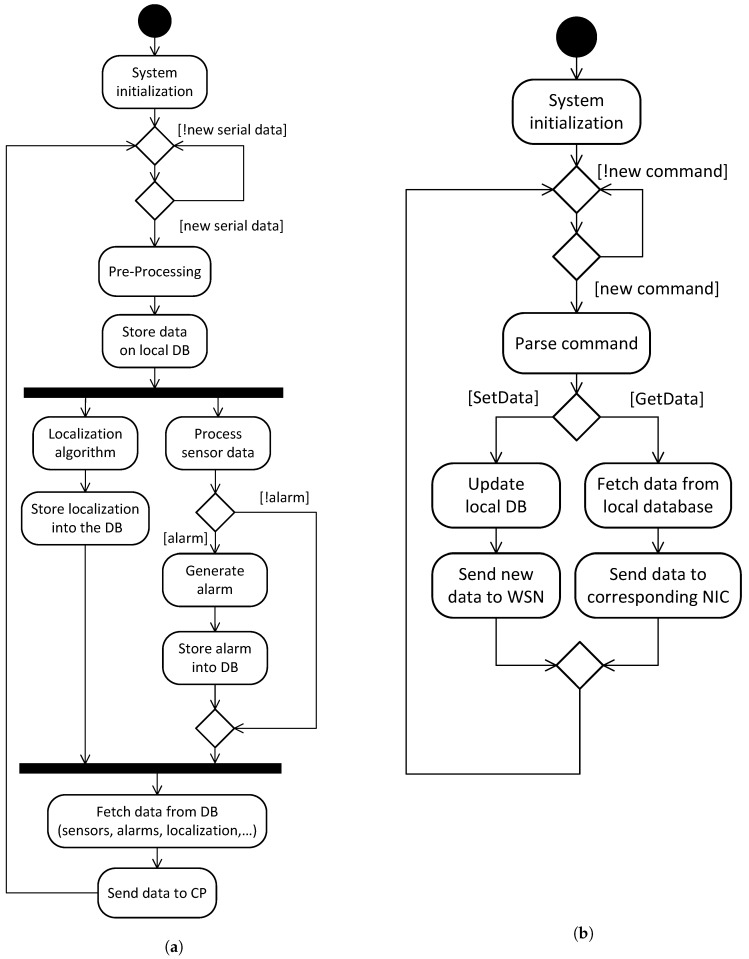
IoT Gateway activity diagrams. (**a**) Localization and alarm procedures. (**b**) Management procedures.

**Figure 10 sensors-19-00603-f010:**
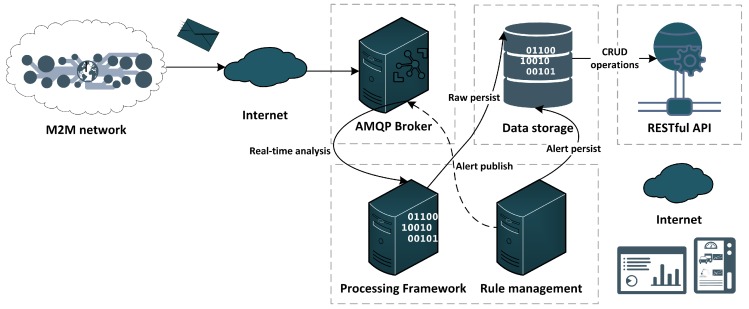
Computational Platform architecture.

**Figure 11 sensors-19-00603-f011:**
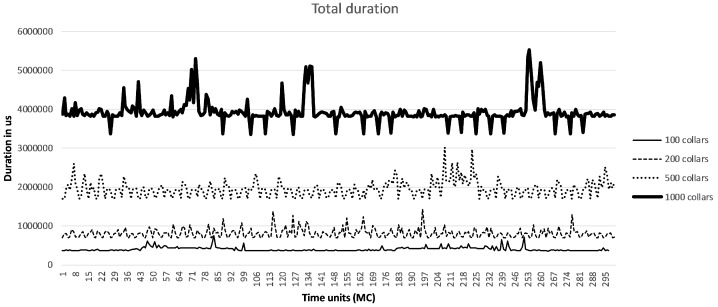
Cumulative processing duration.

**Table 1 sensors-19-00603-t001:** Summary of the most relevant works and commercial platforms on animal monitoring.

Feature/Solution	Localization Monitoring					Activiy Monitoring			
	Nofence	eShepherd	Digitanimal	Huircan	Nadimi	Cowlar	CowScout	Dutta	Alvarenga
	[57]	[73]	[55]	et al. [61]	et al. [67]	[71]	[70]	et al. [65]	et al. [69]
Animals	goats	cattle	several	sheep	sheep	cattle	cattle	cattle	sheep
Data gathering	yes	yes	yes	yes	yes	yes	yes	yes	yes
Real-Time Data	yes	yes	yes	yes	yes	yes	yes	no	no
Localization	GPS	GPS	GPS	RSSI	GPS	no	no	no	no
Virtual fence	yes	yes	no	no	no	no	no	no	no
Activity Monitoring	no	no	no	no	yes	yes	yes	yes	yes
Posture control	no	no	no	no	no	no	no	no	no

**Table 2 sensors-19-00603-t002:** Micro-cycles types currently defined [72].

µC Type—Name	Purpose	MAC Policy
1—Pairing Request (PR)	Device’s pairing	CSMA
2—Collar-to-Beacon (C2B)	Collar (mobile nodes) communications	TMDA
3—Beacon-to-Beacon (B2B)	Inter-beacon relay	TMDA

**Table 3 sensors-19-00603-t003:** The Median Duration of the Most Important Tasks Running on the Gateway.

Task Duration vs. Number of Collars	100	200	500	1000
Duration Rx Beacons (ms)	20.02	20.02	20.01	20.02
Duration Rx Collars (ms)	331.98	661.99	1651.97	3304.96
Duration Process Beacons (ms)	0.49	0.50	0.50	0.50
Duration Process Collars (ms)	2.51	4.92	12.15	24.15
AMQP Library (ms)	23.19	113.15	243.63	492.60
Cumulative time (ms)	380.68	1483.66	1932.25	3859.17

**Table 4 sensors-19-00603-t004:** The worst case measurements (maximum duration measured).

Task Duration vs. Number of Collars	100	200	500	1000
Duration Rx Beacons (ms)	23.132	22.04	22.01	26.82
Duration Rx Collars (ms)	333.25	668.25	1658.69	3306.94
Duration Process Beacons (ms)	2.56	2.79	2.90	3.49
Duration Process Collars (ms)	15.78	26.56	66.05	130.31
AMQP Library (ms)	378.80	742.18	1342.81	2082.95
Cumulative time (ms)	745.84	1430.63	3028.57	5534.48

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
