# Peer review of "An IoT-Based Solution for Intelligent Farming†"

_sensors, 2019, doi:10.3390/s19030603_

Round 1
Reviewer 1 Report
According with the authors, “This paper extends that work (Temprilho et al. [11]) significantly, by improving the state of the art review and presenting a more complete and thorough discussion of the proposed stack. Furthermore, it also details the design, implementation and validation of the IoT gateway, which is the main contribution.”
In my opinion, the state of the art is not good, and the design and implementation of the IoT gateway has a lack of details. See below for details:
Line 41, 42 & 43:
“Even though different approaches can be found in the literature, particularly for IoT and M2M applications, we focus in a generic one, composed of four main layers:”
Which other approaches?
Why the one based on four main layers?
Line 45:
“IEEE.802.15.4” Reference?
Line 46:
A comma needed after IPv6.
Line 53 & 54:
“must be highlighted because of the strength given by Telco companies.”
Justification?
At least, LoRa can be deployed without the participation of any Telco company.
Line 59:
“in a environmentally”
It must be “in an environmentally”
Line 86 & 87:
Twice “the”
Line 73, 74 & 75:
It is not clear that the requirements presented here are related with the communication problem. In general, the paper uses at the same level the concept of communication protocol and IoT platform. And are very different.
Section 2 in general.
The paper use as base “a new M2M protocol stack proposed by Temprilho”. In my opinion, the state of the art should cover: the concept of protocol stack as it is presented in the paper and, in addition, relevant elements of the different levels of the stack. Nevertheless, section 2 (“Related work”) includes an introduction to the existing communications protocols (with special emphasis in LPWAN), as well as to a short view of GPS and how to optimize the use of energy in it. And this GPS review in a section named “Animal monitoring platforms” that almost include only 4 reference to other “platforms” without any detail of functionality, technology used, etc. Under this section (2.4), I hope to find reference to other animal monitoring platforms, and the similarities and differences with SheepIT.
Figure 4 presents the M2M stack, but section 2 does not offer references to the state of the art of the components of most of the elements of Transportation Layer and Application Layer.
Figure 3 is very poor and does not offer any kind of information. More or less the same with figure 5.
Line 318.
Is it a new section? 3.2?
Is it a subsection? 3.1.1? In this case, What is about 3.1.2? It is interesting to have a new subsection if you have additional subsections at the same level.
Regarding with design of the gateway (section 3.2, and part of the new main contribution of the paper):
It is very poor in technical details. No model or diagram based in well-known notation is included. Figure 9 is very poor in technical details and it has been built using general boxes, without any notation that reflect the relationship among components based in time sequence, software engineering concepts, data movement, or any other guiding thread.
There is not any comparison with other solutions provided by other animal monitoring platforms. So, it is impossible to identify its originality and to compare with other approaches, and the advantages of the solution proposed.
In addition, the paper does not explain how the platform can manage the special requirements presented in lines 70 to 78. Section 4 tries to demonstrate the achievement of these requirements, but it does not match them with the capabilities of the system.
Author Response
Before revisiting all the comments and suggestions given, we would like to thank the careful analysis and constructive comments.
We did our best to address all the issues pointed out, being individually tackled. The main changes are *highlighted*on the paper and explained below. There were also further small modifications, namely some English editing and additional references, but that were not highlighted since they do not introduce relevant changes on the paper’s content.
1.1 - Line 41, 42 & 43: Even though different approaches can be found in the literature, particularly for IoT and M2M applications, we focus in a generic one, composed of four main layers:” Which other approaches? Why the one based on four main layers?
Answer: As we were not being clear neither coherent on presenting such issue, we reformulated such paragraph, the fourth paragraph of section 2.1. and the third paragraph of section 3.1.
We can find in the literature several approaches when defining an IoT/M2M stack. For instance, we can find architecture models with 3, 4 or even 5 layers. However, when we move to the deployment of practical solutions, the protocols being used are split into 4 main layers because it aligns with the popular TCP/IP stack (Physical and MAC Layer, Network Layer, Transport Layer, Application layer). Additionally, to tackle the requirements imposed by the SheepIT solution, an IoT architecture based on 4 layers was also considered since it is the one that better fits, allowing a more efficient match between each one of the layers of the proposed stack and system requirements. Consequently, to avoid the dispersion among all the IoT architecture models defined on the literature, we focused only on this one.
1.2 - Line 45: “IEEE.802.15.4” Reference?
Answer: A reference was added.
1.3 - Line 46, 59, 86 and 87
Answer: All those errors were corrected.
1.4 - Line 53 & 54: “must be highlighted because of the strength given by Telco companies.” Justification? At least, LoRa can be deployed without the participation of any Telco company.
Answer: The sentence was reformulated. Although SIGFOX, NB-IoT and LTE-M are in fact being pushed by the strength of Telcos, LoRa can be deployed without the participation of them. Our goal with this sentence was to highlight the increasing popularity of LPWAN solutions.
1.5 - Line 73, 74 & 75: It is not clear that the requirements presented here are related with the communication problem. In general, the paper uses at the same level the concept of communication protocol and IoT platform. And are very different.
Answer: We assume that both concepts were sometimes befuddle along the text. We reviewed all the paper and tried to make that clear. Regarding the requirements, they are in fact directly associated to the needs of the solution, although most of them have direct impact on the communication scheme, for instance:
“Need of real-time localization, preferably resorting on Received Signal Strength Indicator (RSSI)-based localization”: RSSI-based localization mechanisms, besides being attractive in terms of power-consumption and cost, present limited precisions. Thus, in order to reduce the associated error, it is necessary to ensure periodic beacon messages to allow the cross of data and the use of filters to eliminate outliers and/or minimize RF interferences.
“Need to ensure the coexistence of an effective control posture mechanism”: The control posture mechanism needs to monitor sheep’s behaviour (reading the sensors) and apply stimulus (triggering actuators) when undesired behaviors are detected. This demands some processing effort from the microcontroller, that at the same time needs to ensure a correct interaction with the RF interface. Thus, all these tasks must be aligned with the communication tasks in order to avoid loose of efficiency.
“Frequent system reconfigurations by non-technical personnel”: some relevant reconfigurations (regarding the topology of the vine, the intensity of the stimulus, the RF power, etc) shall be allowed to be made on-line, without the need of device’s re-programming. Thus, the communication scheme shall be prepared to allow such interactions.
1.6 - The paper use as base “a new M2M protocol stack proposed by Temprilho”. In my opinion, the state of the art should cover: the concept of protocol stack as it is presented in the paper and, in addition, relevant elements of the different levels of the stack. Nevertheless, section 2 (“Related work”) includes an introduction to the existing communications protocols (with special emphasis in LPWAN), as well as to a short view of GPS and how to optimize the use of energy in it. And this GPS review in a section named “Animal monitoring platforms” that almost include only 4 reference to other “platforms” without any detail of functionality, technology used, etc. Under this section (2.4), I hope to find reference to other animal monitoring platforms, and the similarities and differences with SheepIT.
Answer: We decided to partially reformulate the state of the art organization. First of all, we renamed the subsection 2.1 to “IoT/M2M protocols and communication technologies”. Here, we started by introducing the technologies commonly used on such applications. We consider this relevant because technologies are the basis of any IoT platform. Then, we introduced the most common IoT protocol stack models, focusing on the one we consider more realistic. Having this model as basis, we reviewed the most popular protocols used on each of the layers of the stack. To the end of the subsection, we left the study of the LPWAN solutions and some of Non-IP protocols stacks. We believe that these modifications make much clear this component of the state of the art.
Regarding the animal monitoring platforms, we decided to tackle separately the location monitoring and the behavior monitoring solutions because, usually, they are deployed separately. Concerning the location monitoring, GPS is the main technology used, being that the reason for the focus given. However, we modified that part of the text in order to give more attention to the available platforms instead of focusing on explaining the issues of the GPS. Furthermore, we extended this subsection with more academic research works and commercial solutions. We ended this subsection with a summary table of the most relevant works and platforms regarding the SheepIT requirements.
1.7 - Figure 4 presents the M2M stack, but section 2 does not offer references to the state of the art of the components of most of the elements of Transportation Layer and Application Layer. ???
Answer: The reviewer criticism about the lack of references concerning the Transport and Application Layers is fair. Concerning the Transport Layer, as mostly of applications resort on TCP and UDP that are well known on the network’s domain, we did not paid much attention on their analysis. Even so, we believe that the new organization of the state of the art, turns everything more coherent and clear.
1.8 - Figure 3 is very poor and does not offer any kind of information. More or less the same with figure 5.
Answer: Regarding Figure 3, we consider it important to briefly express an overview of the overall system architecture, without going into to much detail. Notwithstanding, we introduced some relevant changes in order to make it richer and clear, specially concerning the interactions between the devices and system modules.
Figure 5 was included in order to show the types of terrains that we could find on Douro’s region. Nevertheless, we agree that it does not offer any useful information for the present paper, thus it was removed.
1.9 - Line 318. Is it a new section? 3.2?Is it a subsection? 3.1.1? In this case, What is about 3.1.2? It is interesting to have a new subsection if you have additional subsections at the same level.
Answer: In fact, it is a mistake. The text that was following Communication Stack is part of the subsection 3.1. hence, we decided to remove such title.
1.9 - Regarding with design of the gateway (section 3.2, and part of the new main contribution of the paper):
It is very poor in technical details. No model or diagram based in well-known notation is included. Figure 9 is very poor in technical details and it has been built using general boxes, without any notation that reflect the relationship among components based in time sequence, software engineering concepts, data movement, or any other guiding thread.
There is not any comparison with other solutions provided by other animal monitoring platforms. So, it is impossible to identify its originality and to compare with other approaches, and the advantages of the solution proposed.
Answer: The entire subsection 3.2 was totally re-written. Concerning the diagrams, different levels of detail are now given. Firstly, a block diagram exploiting the general hardware modules is depicted and explained. Then, regarding the gateway design, two different fluxograms were newly drawn in order to exemplify the software implementation of different features of the gateway.
Additionally, when discussing the state of the art of existing IoT gateways, we justify the design and implementation of a new gateway, particularly developed for SheepIT and similar intelligent farming solutions.
1.10 - In addition, the paper does not explain how the platform can manage the special requirements presented in lines 70 to 78. Section 4 tries to demonstrate the achievement of these requirements, but it does not match them with the capabilities of the system.
Answer: Section 4.3 discusses not only the results presented on the scope of the text but also how the identified requirements are being tackled by the SheepIT team. Some were already presented on published works, while other still remain to solve. Thus we extended the discussion section in order to address and explain the status of the developments.
Reviewer 2 Report
The paper presents a complete system that should be used for the management of animal herds (e.g. sheep). Several layers are described, with focus on the lower-level layers, especially the MAC layer.
The authors choose to implement a novel solution for IoT communication that relies on two types of nodes (beside the gateway) -- mobile nodes (animal collars) and static beacon nodes. The beacons are fixed and form a mesh network that relays messages from the mobile nodes to the gateway,
While the paper is generally sound, the authors do not discuss with sufficient thoroughness the practical applicability of the proposed protocol in real life. There are two main concerns with the paper.
The first regards the fixed beacons. Considering the actual area that is normally used by grazing sheep, on a period of weeks / months, and the fact that sheep move through multiple areas even during one day's grazing, how realistic is it to use fixed beacons for communication? The experimental setup mentions a 10ha area, but Internet advice mentions about 15ha for 100 sheep. Is it practical to have a large number of fixed beacons? Have the authors consulted a specialist in animal husbandry and livestock?
Second, but related to the first point, the authors do not describe exactly how the experiments were carried out, namely if they were done under realistic conditions, in which nodes are spread on a large area, under bad meteorological conditions, or in the presence of land obstacles (such as hills, mounds, banks, etc).
Finally, although mentioned, the experiments do not illustrate the energy consumption that is implied by the presented setup. Energy consumption is always an important point and data regarding it should definitely be included in the results.
As a smaller comment, the references are poorly edited, they need to me more uniform.
Author Response
Before revisiting all the comments and suggestions given, we would like to thank the careful analysis and constructive comments.
We did our best to address all the issues pointed out, being individually tackled. The main changes are *highlighted*on the paper and explained below. There were also further small modifications, namely some English editing and additional references,, but that were not highlighted since they do not introduce relevant changes on the paper’s content.
2.1 - The first regards the fixed beacons. Considering the actual area that is normally used by grazing sheep, on a period of weeks / months, and the fact that sheep move through multiple areas even during one day's grazing, how realistic is it to use fixed beacons for communication? The experimental setup mentions a 10ha area, but Internet advice mentions about 15ha for 100 sheep. Is it practical to have a large number of fixed beacons? Have the authors consulted a specialist in animal husbandry and livestock?
Answer: In fact, the SheepIT project was developed assuming grazing transhumance, such that ovines could clean different parcels of vineyards (or even complete vineyards) along several days or weeks. Even though this transhumance requires from the operator the re-installation of the fixed beacons every time they need to move the system to a new grazing area, the system was developed in order to simplify such process, being only necessary to place the posts and turn on the devices.
In our perspective, if those beacons would be mobile, for instance placed on sheep, we would face critical issues regarding the autonomy of the devices and/or regarding their efficient operation. On one hand, to maintain the functionalities currently implemented on beacons, we would need bigger batteries than the ones used on collars and, consequently, it would not be viable to have the beacons being transported by sheep. On the other hand, the localization mechanism based on RSSI would present even more limitations since both type of devices would be moving.
The 10ha was chosen taken into account the portuguese reality and considering the practical experience of our partners.
Furthermore, we would like to add that all the tests and conditions were designed considering the inputs given by our project partners namely, the Agrarian School of Viseu and Ramos Pinto, Lda, owner of a vast area of vineyards in Douro’s Region. Consequently, we also added a reference to these partners, particularly when resuming the condition tests.
2.2 - Second, but related to the first point, the authors do not describe exactly how the experiments were carried out, namely if they were done under realistic conditions, in which nodes are spread on a large area, under bad meteorological conditions, or in the presence of land obstacles (such as hills, mounds, banks, etc).
Answer: It order to clarify the experimental evaluation, we extended the introductory explanation of section 4. Particularly, we tried to clarify the experimental evaluations made in previous works in order to contextualize the experiments presented on this paper. In sum, we started by validating the communication mechanism in theory, showing that it meets the solution requirements. Then, on a second work, we transposed that mechanism to a M2M communication stack since we believe it has potential to host other different intelligent farming solutions. Also, we evaluated the implementation of the Physical and MAC layers of such stack. On a third work, we used a specific configuration of the system in order to gather real data from sheep such that we could develop a efficient posture control mechanism resorting on Machine Learning algorithms. Finally, on the present paper, we needed to close the loop, particularly presenting the device not yet introduced, the gateway. The next critical step is, as suggested, the evaluation of the system under realistic conditions. At this moment we are finalizing the development a final prototype such that it could be safely used during larger periods of time on real test conditions.
2.3 - Finally, although mentioned, the experiments do not illustrate the energy consumption that is implied by the presented setup. Energy consumption is always an important point and data regarding it should definitely be included in the results.
Answer: In fact, the energy consumption of the solution is a critical requirement of the system. Nevertheless, along the design of the solution, we focused particularly on collar’s consumption, leaving the beacons and the gateway aside this issue. As beacons and gateways are statically placed in physical posts, we believe that heaveresting energy solutions can be used and thus no relevant issues are expected. The same is not true for collars. We need to ensure a big autonomy in order to avoid constant replacements or recharges of batteries. A theoretical analysis was performed on a previous work and the results are promising (around 3224 hours of autonomy for collars). However, we believe that this value will drastically reduce when tested on real conditions. Firstly, as the sheep behavior is unpredictable, the number of stimulus given by collars can change drastically from sheep to sheep. As the posture control mechanism represents an important source of energy consumption, the autonomy will also differ from collar to collar. Secondly, we believe that the real capacity of batteries do not always reach announced values. Finally, the evaluation considered that the system is running in a steady state, i.e., sheep already passed through a training process that aims at teaching them to react correctly to the posture control algorithm. If not, the training process requires a more reactive system, with the posture control mechanism being run at higher frequencies, thus demanding a higher energy expenditure. To make this clear, we added a new point on the discussion section, summing up this consideration, highlighting that future work is needed on the autonomy’s domain.
2.4 - As a smaller comment, the references are poorly edited, they need to me more uniform.
Answer: We reviewed all the references but we only found a few mistakes. Thus, if you consider that the identified issues remain unsolved, we would appreciate if you could give us one or two examples.
Round 2
Reviewer 1 Report
Congratulations to the authors. They have made a good job in the correctness and improvement of the paper.
Author Response
We would like to thank sincerely to the author for his careful analysis and technical expertise shared. It will certainly contribute to enhancing the future work that it is still to be made on the scope of the project.
Reviewer 2 Report
The paper has been greatly improved and I believe the authors did a good job.
Maybe the authors should mention in the Conclusions section about their intentions of performing experiments in real conditions.
Author Response
We would like to thank the reviewer once more for the contributions given. We checked carefully all paper again, and corrected several small English mistakes/incorrections.